# Advances in the Research of Bioinks Based on Natural Collagen, Polysaccharide and Their Derivatives for Skin 3D Bioprinting

**DOI:** 10.3390/polym12061237

**Published:** 2020-05-29

**Authors:** Jie Xu, Shuangshuang Zheng, Xueyan Hu, Liying Li, Wenfang Li, Roxanne Parungao, Yiwei Wang, Yi Nie, Tianqing Liu, Kedong Song

**Affiliations:** 1State Key Laboratory of Fine Chemicals, Dalian R&D Center for Stem Cell and Tissue Engineering, Dalian University of Technology, Dalian 116024, China; xujie172@mail.dlut.edu.cn (J.X.); Huxueyan@mail.dlut.edu.cn (X.H.); liyingli@mail.dlut.edu.cn (L.L.); liwenfang@mail.dlut.edu.cn (W.L.); 2Zhengzhou Institute of Emerging Industrial Technology, Zhengzhou 450000, China; sszheng@ipezz.ac.cn; 3Burns Research Group, ANZAC Research Institute, University of Sydney, Concord, NSW 2139, Australia; rpar4161@uni.sydney.edu.au (R.P.); yiweiwang@anzac.edu.au (Y.W.); 4Key Laboratory of Green Process and Engineering, Institute of Process Engineering, Chinese Academy of Sciences, Beijing 100190, China

**Keywords:** bioink, skin tissue engineering, 3D bioprinting, wound healing, skin regeneration

## Abstract

The skin plays an important role in protecting the human body, and wound healing must be set in motion immediately following injury or trauma to restore the normal structure and function of skin. The extracellular matrix component of the skin mainly consists of collagen, glycosaminoglycan (GAG), elastin and hyaluronic acid (HA). Recently, natural collagen, polysaccharide and their derivatives such as collagen, gelatin, alginate, chitosan and pectin have been selected as the matrix materials of bioink to construct a functional artificial skin due to their biocompatible and biodegradable properties by 3D bioprinting, which is a revolutionary technology with the potential to transform both research and medical therapeutics. In this review, we outline the current skin bioprinting technologies and the bioink components for skin bioprinting. We also summarize the bioink products practiced in research recently and current challenges to guide future research to develop in a promising direction. While there are challenges regarding currently available skin bioprinting, addressing these issues will facilitate the rapid advancement of 3D skin bioprinting and its ability to mimic the native anatomy and physiology of skin and surrounding tissues in the future.

## 1. Introduction

As the largest organ of the human body, the skin serves as a protective barrier against the external environment, and plays an important role in body temperature regulation, humoral balance, sensory perception, vitamin D synthesis and waste excretion [1]. Skin defects caused by external injuries or diseases often lead to loss of body fluids and bacterial infections, and other life-threatening secondary complications [2]. About 300,000 deaths are annually attributed to burn injuries, while nearly 11 million patients around the world suffer from burns every year. In addition, more than 6 million individuals worldwide suffer from chronic skin ulcers [3,4].

Wound healing involves the complex, highly integrated and overlapping events of hemostasis, inflammation, migration, proliferation and maturation [5,6]. However, damage to skin tissue from high-impact trauma may result in inadequate self-repair and the need for clinical interventions [7]. Current clinical treatments to support wound repair and regeneration include autografts [8], allografts [9], skin substitute [10], cell therapy [11] and cytokine therapy [12]. However, these traditional methods are often limited by the availability of donor skin for grafting, secondary injuries, small repair range, immune rejection, long repair time and high treatment cost [13,14].

Three-dimensional bioprinting, an additive manufacturing technology, was recently introduced and used in the production of cell-laden constructs to renovate the concept of scaffold-based tissue engineering [15,16]. Three-dimensional bioprinting provides a high degree of flexibility and reproducibility, using a computer controlled 3D printer that is capable of fabricating 3D structures through a layer-by-layer printing process [17,18]. Compared to traditional tissue engineering technology, the advantages of 3D bioprinting technology include accurate cell positioning, controllable tissue structure preparation, wide size range and high production capacity [19,20]. In addition, 3D bioprinting has the capacity to promote the formation of vascular structures in tissue engineering, restoring the supply of nutrients and transportation of waste [21]. The spatial accuracy provided by 3D bioprinting has the powerful function of enabling the precise deposition of bioink that will ultimately influence the structural and functional aspects of the bioprinted skin tissue [22].

Bioink, acellular or cell-encapsulating, plays an important role in 3D skin bioprinting [23]. Selecting the appropriate bioink is important as it will influence the overall structure and cellular responses [19,24]. Acellular bioink is mainly composed of biomaterials, while cell-encapsulating bioink also includes living cells and signaling molecules like growth factors [19]. Currently, hydrogel materials (e.g., collagen, gelatin and alginate) are widely used as bioinks in bioprinting skin systems owing to their capacity to encapsulate cells and printability [25,26,27,28,29]. Specifically, collagen hydrogel is commonly utilized for skin repair, because collagen is the most abundant protein-based natural polymer in skin tissue and is a main component of the native extracellular matrix (ECM), which means it is capable of providing a favorable microenvironment [30,31,32]. However, these biomaterials are usually not used alone as a bioink due to the poor mechanical strength and cell adhesion of these biomaterials [33,34,35,36]. Polymer blending and biomaterial composites, however, are of great interest in skin tissue engineering and 3D bioprinting. While there have been advances in skin bioprinting, modelling, vascularization and the auxiliary features remain a challenge for the clinical application of artificial skin [37,38,39]. Therefore, the ultimate goal in skin bioprinting is to engineer fully functional skin that can mimic the native anatomy and physiology of skin and surrounding tissues.

In this review, we summarize the current 3D bioprinting technology for skin tissue engineering, emphasizing the importance of bioink as an important component of 3D skin bioprinting. We discuss the components of bioink, the biomaterials, constituent cells, stem cells and signaling molecules and currently available bioink products for skin bioprinting. The main requirements related to 3D bioprinting for skin regeneration are shown in Figure 1. Finally, we discuss the critical challenges and future approaches in skin bioprinting from a tissue engineering and clinical perspective. Addressing these challenges will facilitate the rapid advancement of 3D skin bioprinting and its ability to mimic the native anatomy and physiology of skin and surrounding tissues.

## 2. Skin Damage and Wound Repair

The skin, the largest organ of the human body, accounting for about 15% of the total body weight in adults, has a very complex multi-layered structure, including epidermis, dermis and hypodermis [39,40,41]. It plays a critical role in maintaining homeostasis, temperature regulation, metabolite transportation, sensory perception and especially acts as an important barrier against the external environment, preventing the invasion of pathogenic microorganisms [42,43,44,45,46]. However, physiopathologic, physical and chemical factors, especially burns, can cause severe skin barrier injuries [47]. Superficial wounds can lead to bacterial invasion and related complications if not treated immediately [48]. Following a disease wound or injury, wound healing occurs, involving the complex, highly integrated and overlapping events of hemostasis, inflammation, migration, proliferation and maturation [5,6,49]. However, in the case of non-healing wounds, the healing process may get obstructed during any of the four stages and gradually become chronic due to failure in healing [50]. Moreover, in conditions associated with severe skin loss (up to 1 m^2^ in severe burn), normal wound healing cannot reconstitute the barrier, leading to high mortality [51].

Current clinical treatments for severe injury include autografts, allografts and xenografts, which mean replacing the damaged skin with the patient’s own skin, a donor’s skin and skin from another species. However, autografts can only be limited to small-scale skin injury due to the limited number of donors as well as the creation of a secondary wound; and allografts and xenografts are possibly related to the risk of immune rejection and disease transmission, except for some ethical and cultural considerations [13,14,52]. In this regard, tissue engineering holds great promise for improving the treatment of skin injuries by providing solutions to challenges that still remain in skin tissue reconstruction, including the restoration of the multilayered native skin architecture and vascular networks [53]. This approach incorporates the use of biomaterials, living cells and signaling molecules to support the formation of functional skin [54]. However, conventional tissue engineering approaches are challenged by the non-homogeneous distribution of cells and the failure to integrate and vascularize upon implantation with the subsequent rejection of the implanted biomaterial along with the formed skin [55]. While some tissue engineered skin products are currently available on the market, there are many limiting factors such as vascularization through the skin substitute, which remains a major and critical limiting factor in clinical success [21,56].

Three-dimensional bioprinting, an additive manufacturing technology, was recently introduced and used in the production of cell-laden constructs to renovate the concept of scaffold-based tissue engineering [15,16]. Three-dimensional bioprinting provides a high degree of flexibility and reproducibility using a computer-controlled 3D printer that is capable of fabricating 3D structures through a layer-by-layer printing process [17,18]. The basic process of 3D bioprinting skin technology firstly involves imaging the wound to construct a bionic structure model (Figure 2) [22]. Three-dimensional modeling and digital image processing technology are then used to gain morphological information on the skin defect, and a stratified skin model is constructed by adaptive stratification methods to generate corresponding printing commands [57]. Skin samples are then taken from the patient to harvest keratinocytes and dermal fibroblasts for subsequent cell sorting and enrichment [58]. The optimal proportion of cell suspension and hydrogel are then used to prepare bioink to print the skin with 3D bioprinting technology. The printed skin is either directly transplanted onto the wound surface or cultured under appropriate conditions to obtain mature skin for transplantation. Hennessy et al. [59] showed that when compared to single-layer skin substitutes, the structure and function of two-layered skin substitutes more closely resembled the structure of native human skin.

## 3. Three-Dimensional Bioprinting

3D bioprinting is a new branch of 3D printing technology used in tissue engineering and regenerative medicine to address the challenges of organ donor shortage [16]. According to different molding principles, current skin 3D bioprinting technology mainly includes inkjet, laser, extrusion, stereolithography and microfluidic bioprinting.

Inkjet bioprinting is a contactless printing that ejects biomaterials and cells from the nozzle in the form of liquid droplets [60] (Figure 3A). Laser bioprinting is a printing technology that irradiates high energy laser pulses onto a thin plate coated with laser absorbing material, and the bubble produced by the bioink on the absorbing layer drives the biomaterial and cells to leave the substrate and deposit onto the platform [61] (Figure 3B). Extrusion bioprinting uses air pressure or a mechanically driven nozzle to extrude bioink and to deposit it on a platform to form a two-dimensional structure. With the movement of the nozzle or the forming platform along the z-axis, the bioink accumulates in a layered manner to form a three-dimensional structure [62] (Figure 3C). Stereolithography bioprinting is a system that uses a projected light source such as a UV bulb or laser to polymerize a polymer solution into a specified pattern [63,64] (Figure 3D). The traditional 3D bioprinting, including laser, inkjet, extrusion and stereolithography bioprinting, are like the reverse process of cutting potatoes—assembling potatoes from mashed potato, diced potato, filar potato and sheet potato, respectively (Figure 3E). Microfluidic bioprinting uses micro-printing equipment based on microfluidic technology, and is different from traditional bioprinters as it is capable of printing artificial skin in a shorter period of time [51,65] (Figure 3Fa,Fb).

### 3.1. Inkjet Bioprinting

Inkjet bioprinting, a drop-on-demand (DOD) printing method, was the first technology used for organ printing [67,68]. Similar to traditional 2D inkjet printing, inkjet bioprinting involves the distribution of bioink in a series of droplets, which are then printed layer by layer to form a 3D structure containing cells using two printing types, piezoelectric type and hot-bubble type (Figure 3A). Hot-bubble printing technology involves heat acting on the bioink near the nozzle to increase the temperature of the bioink and gasification to generate bubbles and force liquid drops to extrude successfully. In piezoelectric inkjet bioprinting, drops are generated by transient pressure waves from piezoelectric actuators. Since inkjet bioprinting follows similar processes, mechanics and functions and only needs to be modified from the general commercial inkjet printer, inkjet bioprinting is the lowest cost bioprinting technology.

An inkjet printing device is simple and cost efficient. Installation of multiple heads on the printer allows for different cells to be printed at the same time and faster printing speeds (1–104 drops/s) compared to other printing technologies. However, due to the small driving pressure of the nozzle, high viscosity materials and high concentration of cells cannot be used, often resulting in the formation of weak skin structures [69]. Furthermore, with hot-bubble and piezoelectric, the threat of mechanical or thermal damage to cells during the bioprinting process must be considered.

### 3.2. Laser Bioprinting

Laser bioprinting is based on the concept of laser-induced forward transfer (LIFT) [70] (Figure 3B). Laser bioprinting involves the use of two layers, an upper glass slide referred to as an energy absorbing layer, and a layer of biomaterials containing cells on the bottom. The absorbing layer receives the pulsed laser, transferring heat to high gas pressure. The hydrogel precursor with cells are then ejected toward the platform. Finally, the 3D structure can be formed with the movement of the platform [71].

Laser bioprinting is different to inkjet bioprinting, having no nozzle and avoiding any direct contact between the bioink and processing device. This non-contact manufacturing method means cells are not exposed to any mechanical stress and can maintain normal cellular activity. Laser bioprinting is compatible with biomaterials of high viscosity, expanding the range of materials that can be used compared to inkjet bioprinting. However, research efforts are still focused on optimizing laser bioprinting, with several reasons limiting its applications including: (a) The high cost of laser-based printers and the availability of commercial printing devices; (b) it is time consuming to apply bioink on laser absorbing material for each printing layer; (c) the repeatability of the resulting droplet needs further study [72].

### 3.3. Extrusion Bioprinting

Extrusion bioprinting evolved from inkjet printing technology and is now a widely used printing technology [69]. According to the operating principle, extrusion bioprinting can be divided into three systems, pneumatic, piston and screw (Figure 3C). The pneumatic system dispenses bioink using compressed gases; piston and screw systems function by using mechanical forces without gases, dispensing bioink through a pump. In the extrusion bioprinting process, continuous extrusion pressure can be used to extrude uninterrupted fibers using various viscosities bioink.

With extrusion bioprinting, a wide range of biomaterials can be used to produce structurally sound skin tissues using a simple printed device that offers low-cost customized services. However, because bioink is extruded by using external mechanical forces that can potentially damage cells, so it needs to reduce the damage as far as possible [73].

### 3.4. Stereolithography Bioprinting

Stereolithography (SLA) has been traditionally used to make cell scaffolds, but has since been used to print bioink with cells. Similar to laser bioprinting, stereolithography bioprinting uses light to selectively crosslink bioink to form three-dimensional structures [74]. Ultraviolet light is selectively projected onto the surface of the bioink using a digital micromirror device before the materials in the irradiated area begin to solidify. Through the up and down movement of the forming platform, a three-dimensional structure is produced by solidification layer by layer (Figure 3D).

The stereolithography bioprinting device uses a digital light projector to cure each layer of bioink with high efficiency. For even complex structural designs, the printing time is the same and the printing accuracy is high [75]. Compared to other 3D bioprinting technologies, the device is simple and easy to control. However, SLA printed 3D structures have been demonstrated to be cytotoxic, reducing the viability of embedded cells [76].

### 3.5. Microfluidic Bioprinting

Microfluidic system is an emerging technology that has been gradually applied in genome sequencing, proteomics, cell biology and medical diagnosis [77]. It has the characteristics of micro, integration, high efficiency, high yield and economy. Due to its integration of cell/tissue culture, biochemical analysis, machine controlled micropump and microvalve, photoelectric reading and wireless micro-control, the microfluidic system is also known as “lab on a chip” (Figure 3Fa,Fb). Leng et al. [51] developed a 3D skin printing device based on microfluidic technology, successfully printing skin tissue with multiple layers under the rotation of a roller.

Microfluidic bioprinting can print a large amount of transplantable artificial skin in a relatively short time, promoting wound repair and regeneration. However, like many 3D bioprinting systems, this technology does not completely model all aspects of native human skin, including hair follicles and pigmentation. The Comparison of the five types of bioprinting techniques is given in Table 1.

## 4. Bioink Components

Three-dimensional bioprinting via a variety of methods involves the rapid stabilization of “ink” upon deposition layer-by-layer to produce a construct intended to interact with cells, tissue or organisms. Bioinks may or may not contain living cells and are referred as cell-encapsulating bioink and acellular bioink, respectively. Three-dimensional bioprinting uses cell-encapsulating bioink to directly print tissues, while acellular bioink (mainly several biomaterials) involves printing scaffolds first and then inoculating cells and signaling molecules (growth factors, cytokines and chemokines) to culture biological tissues.

### 4.1. Biomaterials

In theory, biomaterials used in 3D bioprinting should have the following properties: (1) Printability; (2) biocompatibility and safety; (3) promote cell adhesion and proliferation; (4) conform to the original mechanical properties of skin tissue [86,87]. Biomaterials refer to natural or synthetic macromolecular materials obtained from natural organisms that are biodegradable and have good biocompatibility and low immunogenicity [88,89]. Collagen is commonly utilized in tissue engineering, having physiological properties similar to native skin that can also provide a favorable microenvironment. However, using collagen alone has some disadvantages, lacking in mechanical strength and cell adhesion. To address the limitations of using a biomaterial alone, researchers have blended multi-biomaterials to form composites for skin tissue engineering [51,90,91,92,93]. Here we summarize some commonly used biomaterials for 3D bioprinting. The comparison of crosslinking methods and time of hydrogels commonly used for skin bioprinting are given in Table 2.

#### 4.1.1. Collagen

Collagen is the main component of the skin extracellular matrix (ECM) [94]. It is the most abundant protein in mammals including humans, possessing different cell binding sequences such as Arg-Gly-Asp (RGD) and Gly-Phe-Hyp-Gly-Glu-Arg (GFOGER) that influence adhesion of fibroblasts to the scaffold [95,96]. Collagen based scaffold has also been reported to influence the cellular functions of fibroblasts and keratinocytes, including cell shape, differentiation and migration due to the presence of RGD and GFOGER sequences [97]. It also helps in synthesizing a number of skin ECM proteins to enhance the skin regeneration process. Therefore, collagen hydrogel has been widely used to prepare bioink for skin bioprinting. However, collagen is typically used with other biomaterials because it is too weak to fabricate the scaffolds using the single hydrogel [33].

#### 4.1.2. Gelatin

Gelatin is a protein that is partially hydrolyzed from collagen and contains the Arg-Gly-Asp (RGD)-like sequence, which promotes cell adhesion and migration [98]. Gelatin is derived from collagen but has a higher mechanical strength compared to collagen. In addition, it exerts high hemostatic effect, activates the macrophages to accelerate healing process and does not give rise to antigenicity [29]. Gelatin dissolves only when the temperature is higher than about 40 °C and changes into a gel-like state while it is cooled below 30 °C. The thermal-sensitive property of gelatin means it is a widely versatile polymer in skin bioprinting [34]. However, gelatin is not used as a bioink alone, as it is difficult to optimize the temperature or viscosity of a gelatin solution and undergo a reversible reaction during bioprinting [34].

#### 4.1.3. Alginate

Alginate, a natural polysaccharide derived from seaweed is one of the most widely used polymer for bioprinting applications. Alginate is composed of repeating units of β-d-mannuronic acid and α-l-guluronic acid [99,100]. Alginate had earlier been used for drug delivery and tissue engineering due to its biocompatibility and low cost, but its applications in bioprinting is largely driven by its ability to undergo fast gelation when exposed to Ca^2+^ ions under physiological pH and temperature conditions [99]. However, alginate hydrogel cannot support cell adhesion as it lacks cell adhesive sites, requiring RGD to be incorporated as cell-binding molecules [36]. Therefore, alginate is usually used in combination with other hydrogels for skin bioprinting.

#### 4.1.4. Fibrin

Fibrin is an ECM component of the skin that promotes cell proliferation, differentiation and vascularization [101], and it is spontaneously gelled by the reaction of fibrinogen and thrombin [102]. Cells embedded in fibrin are well dispersed and adhere to proper sites in the printed structure due to the abundance of cell adhesive sites [103]. Therefore, as an initial hydrogel for wound closure, fibrin plays major roles in reducing wound contraction and supporting wound repair. A fibrin-gelatin blended hydrogel has been reported to be used as a bio-paper for skin bioprinting, showing that it provides a natural scaffold for fibroblast embedding and culturing [104].

#### 4.1.5. Hyaluronic Acid

Hyaluronic acid, also called uronic acid, is composed of *D*-glucuronic acid and *N*-acetyl-glucosamine units’ linear polysaccharide [105]. The skin is rich in hyaluronic acid, functioning to regulate the permeability of the blood vessel wall and the diffusion and transportation of electrolytes, to retain water in the skin and to promote wound healing. Hyaluronic acid is a promising polymer as bioink for 3D bioprinting because it is biodegradable, biocompatible and has non-immunogenic properties [106]. However, it is not a stable construct because of its high water solubility [35]. Hakimi et al. [65] printed a three-layer skin tissue sheet that is similar to the structure of natural skin using a bioink composed of alginate, fibrin, collagen and hyaluronic acid by a handheld skin printer for in situ wound healing.

#### 4.1.6. Chitosan

Chitosan, prepared by the deacetylation of natural chitin, is a linear polysaccharide that contains -NH_2_ and -OH active groups that make it easy to combine with other polymers [107]. Chitosan is dissolved in acidic solutions with a pH of 5 or less, as chitosan hydrogel gels with increasing pH solutions [108,109]. Chitosan is a natural biomaterial and its use as a hydrogel for wound dressing and skin tissue engineering has been previously investigated. Chitosan has good antibacterial properties that also makes it an attractive biomaterial for the production of artificial skin. Yang et al. [92] prepared a bioink composed of chitosan, hyaluronic acid and collagen with tyrosine for skin scaffold bioprinting. The results showed that the skin substitute made of the bioink were mechanically strong and the substitute is promising in skin repair.

#### 4.1.7. Pectin

Pectin, a branched chain polysaccharide extracted from the cell wall of plants, has gained attention in the tissue engineering field due to its biocompatibility, biodegradability and ability to form physical hydrogels through interaction with calcium ions as it has blocks of galacturonic acid residues [110,111]. Otherwise, multiple polysaccharide domains of pectin provide a variety of target sites for chemical modification and bioactive properties involved in bio-adhesivity, anti-cancerous effects and immunomodulatory activity, which is useful in tissue engineering [112]. A feature of pectin that distinguishes it from other biomaterials is the lack of endogenous cell-adhesive and cell-proteolytic sites, which provides the possibility to introduce a biochemical molecule onto the bioinert polymer backbone to reduce their effect on cell behavior and improve cell viability [113]. Pereira et al. [114] prepared a pectin hydrogel using cell-degradable peptide crosslinkers and integrin-specific adhesive ligands. The results demonstrated that the pectin hydrogels support the in vitro formation of full-thickness skin and are thus a highly promising platform for skin tissue engineering application.

Three-dimensional printing of complex tissue and organ systems will require a variety of biomaterials to model different microenvironments and to support cell growth. However, many biomaterials, when used alone, have low mechanical strength and cannot promote cell adhesion and cell growth, making them not ideal biomaterials for bioink. Therefore, bioink used for 3D bioprinting is composed of multiple biomaterials. In future the aim would be to develop a class of bioink that is not only distinct in composition and biological functionality, but can also be tailored to have very similar printing and post-processing requirements [115].

### 4.2. Cells

Human skin mainly consists of fibroblast, keratinocytes, melanocytes, Langerhans cells, Merkel cells, adipocyte and immune cells. To successfully 3D bioprint human skin, a range of cells must be incorporated and can be amplified. The source of seed cells can be divided into three categories: (1) Primary cells of human tissue; (2) human stem cells; (3) foreign cells. While autologous cells can eliminate graft rejection, the use of allogeneic/heterogenous cells are more likely to induce immune rejection, disease transmission and high recurrence rate [59]. Therefore, autologous cells are commonly used for 3D bioprinting.

Human primary cells, which typically include keratinocytes, fibroblasts and melanocytes, are usually obtained from human foreskin tissue. Keratinocytes, the main component of the epidermis, can be isolated and cultured in the laboratory. Fibroblasts have a paracrine effect on keratinocytes, enhancing the uniformity of the formed stratum corneum. Stem cells are capable of self-renewal and differentiation. Stem cells, such as epidermal stem cells, hair follicle stem cells, adipose-derived stem cells, and mesenchymal stem cells, are an important source of seed cells for skin 3D bioprinting, providing progenitor cells for the development of blood vessel, hair follicles and sweat glands [117]. Bioprinted amniotic fluid-derived stem cells and mesenchymal stem cells resuspended in a fibrin-collagen hydrogel have been reported to accelerate healing of large skin wounds [94]. However, stem cell differentiation is non-directional [118], and printed tissues may face the formation of malignant malformations and long-term adverse effects [119].

Unlike stem cells, progenitor cells have a limited number of divisions and represent intermediate cells that are committed to the differentiation of a target cell [120]. Ou et al. [121] used 3D bioprinting technology to print the dermis and epidermis successively. Sweat gland and hair follicle progenitor cells were implanted into the dermis, which resulted in the production of full-layered skin with sweat glands and hair follicles, accelerating skin healing. Huang [82] obtained a 3D structure that induces differentiation of epidermal progenitor by incorporating hydrogels, dermal homogenates and epidermal growth factor. Moreover, direct delivery of bioprinted 3D structure into burned paws of mice resulted in functional restoration of sweat glands.

### 4.3. Growth Factors, Cytokines and Chemokines

Wound healing is an evolutionarily conserved, complex and multicellular process that aims to restore structural and functional integrity of skin. This process involves migration, infiltration, proliferation and differentiation of several cell types including keratinocytes, fibroblasts, endothelial cells, macrophages and platelets, culminating in an inflammatory response, the formation of new tissue and wound closure. The complex process is executed and regulated by an equally complex signaling network involving numerous growth factors, cytokines and chemokines. Major growth factors, cytokines and chemokines that are involved in the wound healing process are shown in Table 3.

Growth factor, a transmit signal that influences cell activity, stimulates or inhibits cellular proliferation, differentiation and ECM secretion [122]. Therefore, growth factor plays a significant role in the wound healing process [123] There are many types of growth factors, including the epidermal growth factor (EGF) family, fibroblast growth factor (FGF) family, transforming growth factor-β (TGF-β) family, platelet derived growth factor (PDGF), vascular endothelial growth factor (VEGF), connective tissue growth factor (CTGF) and granulocyte macrophage-colony stimulating factor (GM-CSF).

The role of the EGF family, including EGF, heparin binding EGF (HB-EGF), transforming growth factor-α (TGF-α), epiregulin, amphiregulin, betacellulin and neuregulin, has been well characterized in the wound healing process [124]. Almost of them are secreted by platelets, macrophages and fibroblasts, and act on keratinocytes through increasing cell proliferation and migration, thus promoting reepithelialization [125]. Otherwise, the HB-EGF is secreted by keratinocytes and works in an autocrine fashion also promoting angiogenesis [126]. The FGF family is composed of 23 members, of which the most important members are FGF-2, FGF-7 and FGF-10. The FGFs are secreted by keratinocytes, fibroblasts, endothelial cells, smooth muscle cells, chondrocytes and mast cells [127]. Then FGFs bind and activate the FGF receptors to promote the migration of keratinocyte and fibroblasts, and stimulate these cells to produce various ECM components, which play an important role in granulation tissue formation, tissue remodeling and reepithelialization [128,129]. The TGF-β family includes TGF-β1-3, bone morphogenic proteins (BMP) and activins. Among them, the role of TGF-β1 predominates in wound healing. The TGF-β are secreted by keratinocytes, fibroblasts, macrophages and platelets when the skin is injured, and facilitate the recruitment of inflammatory cells, augment tissue debridement of macrophage and initiate expression of genes associated with ECM formation, promoting connective tissue regeneration, tissue remodeling and reepithelialization [130,131]. In addition, TGF-β1 is also involved in up-regulating of VEGF [132]. Therefore, TGF-β1 also plays a role in angiogenesis during wound healing. The PDGF family is comprised of PDGF-AA, PDGF-AB, PDGFBB, PDGF-CC and PDGF-DD. They are secreted by platelets, macrophages, keratinocytes, fibroblasts and vascular endothelium cells, and stimulate the chemotaxis of inflammatory cells to the wound site, augment tissue debridement of macrophage and granulation tissue formation and recruit pericytes to the capillaries, thus enhancing blood vessel maturation during angiogenesis [133]. In addition, PDGF can also promote the proliferation of fibroblasts and thus reepithelialization [134]. The VEGF family, including VEGF-A, VEGF-B, VEGF-C, VEGF-D, VEGF-E and placenta growth factor (PLGF), are widely secreted by all stages of cells during wound healing, enhance migration and proliferation of the endothelial cell in the early stage of angiogenesis and stimulate inflammatory and hematopoietic cell recruitment to the wound site, promoting inflammation and angiogenesis [135]. CTGF, an ECM-associated heparin-binding protein, is synthesized by fibroblasts and promotes proliferation and chemotaxis of these cells, and is involved in the formation of granulation tissue, tissue remodeling and reepithelialization [136]. GM-CSF is increasingly secreted by keratinocytes in the inflammatory stage of wound healing, works on neutrophils to promote their function in the wound site and acts on keratinocytes and endothelial cells to enhance their proliferation and migration, thus promoting reepithelialization and angiogenesis [137].

Cytokines, particularly interleukin-1 (IL-1), IL-6 and TNF-α, play an important role in promoting inflammation. IL-1 is immediately released by keratinocytes during wound healing. It has a paracrine and autocrine effect, increasing keratinocyte proliferation and migration, and activates fibroblasts to secrete FGF-7 [138]. IL-6 is produced by neutrophils and monocytes and initiates kick-starting the healing response both via its mitogenic effects on keratinocytes and its chemoattractive effect on neutrophils during the inflammatory phase of wound healing [128]. Similar to IL-1, TNF-α induce the production of FGF-7 to indirectly promote reepithelialization. However, it shown that TNF-α promotes wound healing at low levels, but has detrimental effect on healing at higher levels [128].

Chemokines (chemotactic cytokines), primarily α-(CXC-) and β-(CC-), are a kind of small cytokine that contribute to the migration of inflammatory cells in the wound site and regulation of tissue remodeling, angiogenesis and reepithelialization. α-(CXC-) chemokines are secreted by keratinocytes, fibroblasts and endothelial cells; then they recruit inflammatory cells to the wound site, thus promoting angiogenesis, and enhance proliferation and migration of keratinocytes and endothelial cells, thus contributing to tissue remodeling and reepithelialization [139,140]. β-(CC-) chemokine is induced in keratinocytes, and acts as a chemoattractant for inflammatory cells, contributing to prolong the inflammatory response [141].

## 5. Classification of Bioink

### 5.1. Acellular Bioink

Acellular bioink offers numerous advantages in skin tissue bioprinting. Acellular bioinks do not necessarily need to be cell-compatible prior to or during 3D bioprinting until cell seeding. For example, bioinks or structures may contain organic solvents for crosslinking as long as they are effectively removed before application of cells or implantation [142]. In addition, high temperatures (>60 °C) utilized for extrusion and physical or chemical means such as heat, light, or plasma utilized for sterilization may also be employed during post-bioprinting processing steps. Moreover, acellular bioinks can also be synthesized and stored prior to printing, resulting in printed constructs that can be stored and packaged for later use [143]. Furthermore, acellular bioinks allow for the production of constructs with more distinct and functional properties compared to those constructed from cell-encapsulating bioink, due to the range of materials and processing options available. Therefore, the bioprinting tissue model with acellular bioink has long been the common method despite the relatively complexity operation.

### 5.2. Cell-Encapsulating Bioink

Cell-encapsulating bioink, which are comprised of biomaterials and living cells as well as growth factors, cytokines and chemokines, must conform to a rigorous set of cell compatibility requirements. Cell-encapsulating bioinks are also likely to be chemically or physically crosslinked to form hydrogels to provide encapsulated cells with an appropriate environment to support the passive diffusion of nutrients and waste to sustain cell viability [144]. However, crosslinking methods must be noncytotoxic, since they is performed in the presence of the encapsulated cells. In addition to facilitating adequate nutrient and waste diffusion, the polymer network must also support cell proliferation and migration [145]. Shear forces applied to the bioink during extrusion through a fine diameter nozzle must be favorable to cell viability, and ensure that any chemical or physical changes can be tolerated by the cells [146]. Throughout the entire fabrication process, sterility must be maintained to preserve cell viability. Furthermore, an often overlooked but important factor when working with cell-encapsulating bioink is to prevent cell settling as much as possible, due to variable cell densities, particularly those of low viscosity [19,147,148]. Therefore, formulating a cell-encapsulating bioink with an appropriate viscosity also remains a major challenge.

## 6. Current Bioink Products for Skin Bioprinting

A number of bioink products are currently available, including ionic crosslinked, thermo-sensitive, photosensitive and shear-thinning bioink (Figure 4). The decellularized extracellular matrix (dECM) bioink has also emerged and has many investigating its potential applications. The bioink products for skin bioprinting research are given in Table 4.

### 6.1. Ionic Crosslinked Bioink

As the name suggests, ionic crosslinking bioink performs the solidification of hydrogels via an ionic crosslinking reaction (Figure 4A). For example, alginate series ink, the sodium ions in sodium alginate are replaced with calcium ions to obtain calcium alginate hydrogel. As shown in Table 4, the ionic crosslinked bioink was suitable for extrusion, inkjet, laser and microfluidic bioprinting.

A hydrogel structure composed of alginate, cellulose nanofiber and fibrinogen has been successfully produced using inkjet-spray printing, a new bioprinting process that combines drop-on-demand inkjet printing with a spray-coating technique (Figure 5A) [93]. This printed skin tissue construct closely mimicks the structure of native tissue (Figure 5B). Further research has also reported that the incorporation of honey as a bioink component can improve the biological properties of an alginate-based bioprinted construct [149]. A novel dermal substitute scaffold with sodium alginate/gelatin composite material as bioink via extrusion technique and three-stage crosslinking process has also been manufactured [29]. Kim [150] used a mixture of collagen and alginate as a bioink to print scaffold, and then used EDC and CaCl_2_ solutions for crosslinking. After freeze-drying, keratinocytes and fibroblasts were inoculated successively for medium immersion culture and air–liquid culture. In vitro tests showed that cells could migrate and proliferate on the stable collagen–alginate scaffold, forming a dense structure similar to human skin (Figure 5C). This scaffold construct was also grafted on the dorsal area of mice, and was found to promote neovascularization and the formation of new granulation tissue (Figure 5D). Huang et al. [82] combined gelatin and sodium alginate, with a proportion of dermal homogenates and endothelial progenitors as well as epidermal growth factor to prepare a bioink for bioprinting via extrusion bioprinting. Fluorescence detection after 24 h of culture showed that the cell activity was high (Figure 5E), and the wound healed well after 14 days transplanted into the mice foot skin (Figure 5F). This provides extensive research thought for the construction of 3D printed extracellular matrix skin.

### 6.2. Thermo-Sensitive Bioink

Thermo-sensitive bioink performs the transformation from sol to gel state by heating or cooling (Figure 4B). For example, gelatin based bioink, requires a heating nozzle to melt gelatin during the printing process, followed by cooling on the printing platform to allow for gelatinization. As shown in Table 4, the thermo-sensitive bioink was suitable for extrusion and inkjet bioprinting.

A newly developed bioink composed of gelatin, alginate, fibrinogen and fibroblasts has been reported to closely model the structure of human skin at the macroscopic and molecular level [152] (Figure 6A,B). Similarly, a fibrin-gelatin hybrid bioink has also been prepared to fabricate a hydrogel with inoculated fibroblasts for skin bioprinting [104]. Furthermore, 3D skin structures have also been printed using a collagen/gelatin hydrogel, where the gelatin pattern was printed into the collagen groove under 20 °C, and the gelatin was selectively liquefied and removed under 40 °C in an incubator to form a fluid channel scaffold (Figure 6C) [153]. The results demonstrated that, compared to skin scaffolds without fluid channels, skin scaffolds with fluid channels had significantly better cell viability (Figure 6D). Ionic crosslinked bioink has also been combined with a thermo-sensitive bioink. For example, a mixed hydrogel composed of gelatin and alginate was casted around the wound mold with Ca^2+^ ionic and placed in 4 °C for gelation (Figure 6E) before printing a skin graft on the burn wound bed by in situ bioprinting [27].

### 6.3. Photosensitive Bioink

Photosensitive bioink performs the transformation from sol to gel state by the activation of a light initiator in the bioink (Figure 4C). As shown in Table 4, the photosensitive bioink was suitable for stereolithography, extrusion, and inkjet bioprinting. For the bioprinting of dermal constructs a single-component hydrogel bioink made from pectin, modified with methacrylate and a controlled density of cell-adhesive ligands has been previously produced [156]. The methacrylate modified pectin bioink was designed to allow the tethering of integrin-binding motifs and the formation of hydrogels by UV photopolymerization and ionic gelation. Under optimal conditions, this bioink can print a porous 3D construct with high structural fidelity to support the growth of fibroblasts (Figure 7A) and facilitate the secretion of extracellular matrix (ECM) proteins including collagen, fibronectin and laminin. Further research has also included the use of protease-degradable pectin hydrogels, obtained by photoinitiated thiol-norbornene click chemistry (Figure 7B), and exhibiting tunable properties that are capable of modulating the behavior of embedded fibroblasts. The inoculation of keratinocytes using these protease-degradable pectin hydrogels has also allowed for bioprinted skin constructs that more closely resemble native human tissue [114]. Silk fibroin (SF)-incorporated four-arm polyethylene glycol acrylate (PEG4A) has been reported to be used as a bioink for digital light processing 3D bioprinting [157]. The printed hydrogel supported cell proliferation and spreading, following the formation of a thicker keratin layer. A photosensitive bioink composed of polyethylene glycol diacrylate (PEGDA) and methacrylate gelatin (GelMA), with visible light crosslinking has been used for artificial skin tissue printing using a low-cost stereolithography bioprinter (Figure 7C) [158]. NIH3T3 fibroblast cells test showed that the system presented a good cell viability for five days of culture (Figure 7D).

### 6.4. Shear-Thinning Bioink

Shear-thinning bioink exploits the phenomenon of apparent viscosity of material decreases with increasing shear stress. When materials are not subjected to any shear force these materials remain in a gel state; however, when subjected to shear force these materials convert to a sol state (Figure 4D). As shown in Table 4, the shear-thinning bioink was suitable for extrusion, inkjet, laser and microfluidic bioprinting.

Collagen is the most abundant protein in the dermis and is widely used in skin tissue engineering. As such, the combination of collagen, fibroblasts and keratinocytes has been used as bioink for laser bioprinting of human skin structures (Figure 8A) [83]. For hybrid 3D cell-printing the use of extrusion and inkjet modules at the same time have been proposed [160]. Collagen with polycaprolactone (PCL) mesh, to prevent its contraction during tissue maturation, and fibroblasts have been reportedly used as bioink to print dermis by extrusion bioprinting on a polydimethylsiloxane (PDMS)-based microfluidic device. At the same time, keratinocytes were printed by inkjet bioprinting, where keratinocytes were evenly distributed on the dermis to produce a multilayered skin structure. (Figure 8B). The results show that the bioprinted structure supports the growth and protein secretion of the cells (Figure 8C). Similarly, a full-thickness skin model has been produced by using a collagen-based bioink with fibroblasts for dermis printing, followed by subsequent air–liquid interface culture for the inoculation of keratinocytes and melanocytes [79]. A collagen-based bioink to fabricate 3D biomimetic hierarchical porous structures by two-step drop-on-demand bioprinting has allowed for successful skin bioprinting [116]. Compared to traditional methods, this 3D bioprinted skin had uniform cell distribution, distinct layering, uniform coloring, good biological performance and revealed a high degree of similarity to human skin morphology (Figure 8D).

An artificial skin with good mechanical properties was printed using a composite bioink consisting of gelatin methyl acrylamide (GelMA) and collagen (Col), with added tyrosine (Ty) to crosslink GelMA and Col. The result showed that Ty can enhance the proliferation of human melanocyte and migration of human dermal fibroblast, and the artificial skin can facilitate wound healing and prevent scarring [91]. Similarly, a bioink composed of chitosan, hyaluronic acid and collagen with tyrosine, to enhance the mechanical strength, has been developed for 3D bioprinting of a skin scaffold [92]. A collagen-based bioink was printed and cultured at the air–liquid-interface (ALI) for one day before transplanting it onto the superficial skin in mice [161]. The results showed that a significant amount of E-cadherin, angiogenesis and secreting collagen were presented in a thin layered structure that conformed to the structure of human skin after 11 days of culture (Figure 8E).

### 6.5. Decellularized Extracellular Matrix (dECM) Bioink

To prepare decellularized extracellular matrix (dECM) bioink, extracellular matrix from native animal (such as porcine) tissues undergoes decellularization and solubilization processes. After the decellularization process, a majority of immunogenic components such as cells and DNA are removed. The extracellular matrix components such as collagen, glycosaminoglycans (GAGs), hyaluronic acid and elastin are largely retained, and are the main components of the tissue microenvironment to support cell growth, adhesion, proliferation, migration and differentiation. The skin-derived decellularized extracellular matrix (S-dECM) therefore represents a promising source for bioink formulation in 3D bioprinting skin tissue engineering. As shown in Table 4, the dECM bioink was suitable for extrusion and inkjet bioprinting.

Porcine S-dECM bioink has been reported to be obtained by decellularizing, and retention of ECM components, including collagen, glycosaminoglycans (GAGs) and hyaluronic acid (HA) [122] (Figure 9A). In vitro testing by the inoculation of human dermal fibroblasts (HDF) and human epidermal keratinocytes (HEK) demonstrated that the S-dECM provides a stable tissue-specific microenvironment for the production of artificial skin, promoting epidermal organization, dermal ECM secretion, and barrier function (Figure 9B). Furthermore, human adipose mesenchymal stem cells (ASCs) and endothelial progenitor cells (EPCs) were added to the printed skin for in vivo testing. The results showed that the addition of ASCs and EPCs accelerated wound closure and reepithelization (Figure 9C) as well as neovascularization. A new 3D cell printing system has been established using dECM as the bioink, which completely gelatinizes at 37 °C, and equipped with a heating module allowing for temperature adjustments to precisely stack bioink with cells (Figure 9D) [23]. The results show that the heating system does not have negative effects on cell viability and activity (Figure 9E). Many researchers have also investigated the use of the non-skin tissue decellularized extracellular matrix for skin bioprinting, including potential applications with the use of decellularized small intestinal submucosa [81], and human acellular amniotic membrane (HAAM) [163].

## 7. Discussion and Future Perspective

To assess the most suitable bioink for skin bioprinting, bioinks are generally evaluated by their printability, biocompatibility, mechanical properties and degradability. Printability is used to evaluate the forming performance of the bioink, requiring the viscosity of bioink to be adjustable and controllable, fast transformation speed from sol to gel state and wide printing process parameters. Biocompatibility is used to evaluate the ability of bioink to simulate the extracellular matrix of native human skin, which requires the printed structure to closely resemble the microenvironment of cells in the body. The printed bioink scaffold is expected to support normal cellular function, and promote cell proliferation, migration and differentiation [164]. After gelation, the mechanical properties of the bioink scaffold are expected to support any processes prior to implantation in the body. Furthermore, the printed structure must be degraded, absorbed and metabolized when new tissue is formed. The degradation rate of the skin scaffold should match the rate of new skin formation to meet the conditions required for skin regeneration.

The extracellular matrix secreted by human cells are mainly composed of structural protein such as collagen, elastin, proteoglycans such as glycosaminoglycans (GAGs), and specific protein such as osteonectin, tenascin and fibrin, which only present in certain tissue. The ideal bioink is expected to closely model the natural extracellular matrix, and depending on the different cells required for tissue bioprinting, the bioink should be adjusted to support specific cell growth and activity. For example, when printing cartilage cells, the addition of hyaluronic acid (common component in cartilage) will promote the proliferation and differentiation of chondrocytes [165]. In selecting the appropriate bioink for skin bioprinting, it is therefore important to select a bioink that is similar to the natural microenvironment of keratinocytes, fibroblasts and melanophores, that will support their survival, adhesion, proliferation, migration and differentiation.

Currently, hydrogel materials such as collagen, gelatin and alginate are widely used as bioink in bioprinting skin systems owing to their capacity for cell encapsulation and printability [25,26,27,28,29]. Specifically, collagen hydrogel is commonly utilized for skin repair, because collagen is the most abundant protein-based natural polymer in skin tissue and is a main component of the native extracellular matrix (ECM) [30,31,32]. However, the use of collagen alone is limited by poor mechanical strength and slow forming speed, requiring subsequent modification by combining it with other biomaterials. As gelatin dissolves when the temperature is higher than 40 °C and gelates below 30 °C, gelatin is a biomaterial that can be used as a thermo-sensitive bioink. However, as gelation is a reversible reaction, using gelatin alone for skin bioprinting would not be stable [34]. Alternatively, alginate hydrogel can undergo fast gelation under exposure to Ca^2+^ ions, and has good mechanical properties that make it suitable for fabricating cell-laden tissue constructs. Nevertheless, its weak biocompatibility, lack of cell adhesion sites and free Ca^2+^ ions may affect cell viability and proliferation [36]. Other commonly used biomaterials such as hyaluronic acid, fibrin, pectin and chitosan have similar shortcomings when used alone for bioprinting. Therefore, bioink used for 3D bioprinting skin is composed of multiple biomaterials.

Polymer blending is of great interest in skin tissue engineering, allowing more control in the development of biomaterials with properties to match those of native skin [78]. As a strategy to overcome the limitations associated with pure polymeric systems, the incorporation of other materials provides an opportunity to build and enhance characteristics of the composite as a whole. For example, alginate has been previously blended with gelatin as alginate gelates easily in the presence of Ca^2+^ and has a good mechanical strength, and gelatin offers RGD sequences for cell adhesion and migration [29]. Biomaterials that are commonly blended also include collagen blended with nanofibrous [80], alginate blended with honey [149], gelatin blended with fibrin [104] and methacrylate gelatin blended with polyethylene glycol [158]. Crosslinking of the blended bioink, through thermal, chemical and photo-crosslinking, is critical for the maintenance of the 3D structure that provides the proper mechanical properties and microenvironment for cellular activities [166]. However, harmful crosslinking reagents and photoinitiators for hydrogel gelation may cause poor biocompatibility and high cytotoxicity, owing to the residues produced after crosslinking [167]. Therefore, it is very important to control the concentration of crosslinking reagent and crosslinking degree to reduce the damage to cells during and post bioprinting process. For instance, the effects of calcium ion, photoinitiator concentration and crosslinking time on cell activity have been previously investigated for skin bioink and bioprinting [156]. Furthermore, a 3D cell printing system has been developed with heating modules for the simultaneous crosslinking of dECM bioink, which is simple and safe for cellular activities compared to other bioprinting processes [23].

While there have been recent advances in skin bioprinting, several barriers still remain that limit the clinical application of 3D bioprinted artificial skin. The most critical challenge is the need for large skin construct with highly developed vasculature [38]. Blood vessels are critical for cellular activity, later tissue maturation and long term potency of the printed tissue after implantation. Reducing the period between implantation and angiogenesis is a critical challenge and greatly influences the success of any graft implanted in the body. A remaining challenge also involves the printing of a multi-layered and intact complex skin structure composed of epidermal, dermal and hypodermal components [38]. It is important to use the printed intact artificial skin to match with the native skin of the patient in the wound to facilitate the wound healing. Furthermore, as wound healing involves the functional recovery of skin structures, bioprinted skin are also expected to support the development of sweat glands, hair follicles, sebaceous glands, sensory nerves and melanin pigmentation. In the future, it is critical to engineer fully functional skin by bioprinting structures that closely mimic the native anatomy and physiology of skin and surrounding tissue.

There are many factors that should be considered in 3D skin bioprinting, including different bioprinting strategies, biomaterials, cells, growth factors, drugs and other additives. Innovations in any of these areas will drive the development of skin tissue engineering. For example, integration of biosensor technology with bioprinting may generate skin on a chip, which has immense potential in the study of pathophysiology of skin defects and drug screening for skin diseases. Such systems can simulate inflammation and edema that will prove useful when testing drug-based treatments [168]. To date, there are a wide range of available biomaterials for use in 3D skin bioprinting. Recently, honey [149] and pectin [114] were used for the first time in skin bioprinting and were reported to produce artificial skin with good mechanical properties and biocompatibility. Despite advancements with the use of skin cells (keratinocytes, fibroblasts and melanocytes) and multipotent stem cells (ADSCs, MSCs) in bioprinting of skin tissue constructs, the use of induced pluripotent stem cells (iPSCs) may provide more opportunities in skin bioprinting [169,170]. Moreover, advancements in skin bioprinting may also come with the co-culturing of various cells that are usually found in the skin layers and related tissues including keratinocytes, fibroblasts, pericytes, neural cells and ligament cells. Furthermore, future research may also attempt to develop specific bioprinted constructs with multifunction involving antibacterial, anti-inflammatory and antiviral properties in addition to the existing features involving sebaceous glands, sweat glands, hair follicles, melanin pigmentation and sensory nerves for wound healing.

In summary, by combining state of the art tissue engineering strategies and achievements made by current and ongoing research, there have been promising advancements towards the development of fully functional bioprinted skin. By integrating multiple disciplines such as bioengineering, cell biology, material science and chemistry, the groundwork has been laid to propel the development of ideal 3D bioprinting artificial skin to successfully translate into clinical applications in the future.

## 8. Conclusions

Three-dimensional skin bioprinting, through inkjet, laser, extrusion, stereolithography and microfluidic technology, is an important processing technique in the field of skin tissue engineering. It provides a high degree of flexibility and reproducibility using a computer-controlled or a mini microfluidic 3D bioprinter to fabricate skin structures via a layer-by-layer printing process. Acellular and cell-encapsulating bioinks play a critical role in the skin bioprinting process. Acellular bioink mainly contains biomaterials, while cell-encapsulating bioink also includes living cells, growth factors, cytokines and chemokines. There are wide ranges of biomaterials that can be used for skin bioprinting bioink, and the biomaterials are generally not used alone as bioinks due to limitations in mechanical strength and lack of cell adhesion. Polymer blending offers the opportunity to build and enhance characteristics of polymer composites as a whole. While there have been advancements in skin bioprinting, several challenges, including vascularization, the development of multi-layered intact artificial skin and promoting the development of auxiliary structures, limit potential clinical applications. In the future, it is critical to engineer fully functional skin by bioprinting structures that closely mimic the native anatomy and physiology of skin and surrounding tissue.

## Figures and Tables

**Figure 1 polymers-12-01237-f001:**
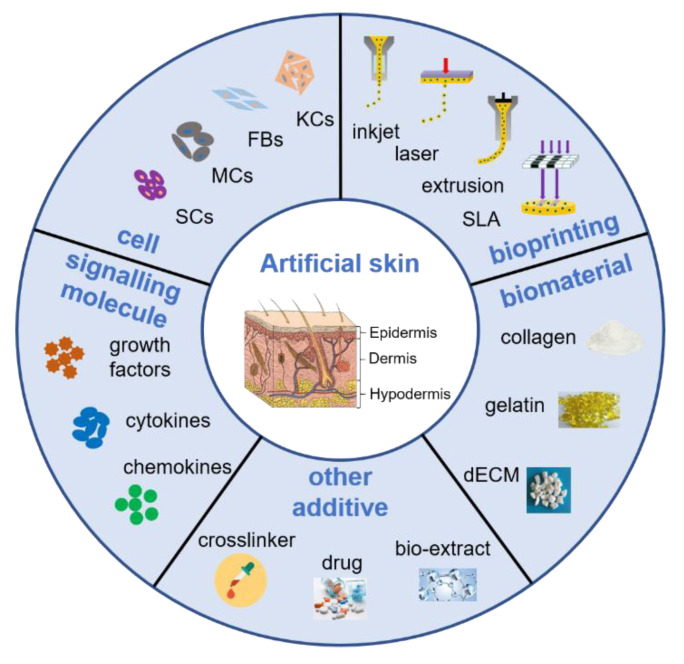
Schematic representation of the main requirements related to 3D bioprinting for skin regeneration. The artificial skin was printed by inkjet, laser, extrusion or stereolithography (SLA) bioprinting technologies with a cell-encapsulating bioink, which consists of biomaterials, constituent cells, stem cells and signaling molecules, or acellular bioink which contains biomaterials only. The mechanical property of artificial skin was enhanced by adding crosslinker, and some drugs or bio-extract were adding to obtain multifunctional skin for wound healing, such as anti-inflammatory and antibacterial. Abbreviations: KCs: Keratinocytes, FBs: Fibroblasts; MCs: Melanocytes, SCs: Stem cells, dECM: Decellularized extracellular matrix.

**Figure 2 polymers-12-01237-f002:**
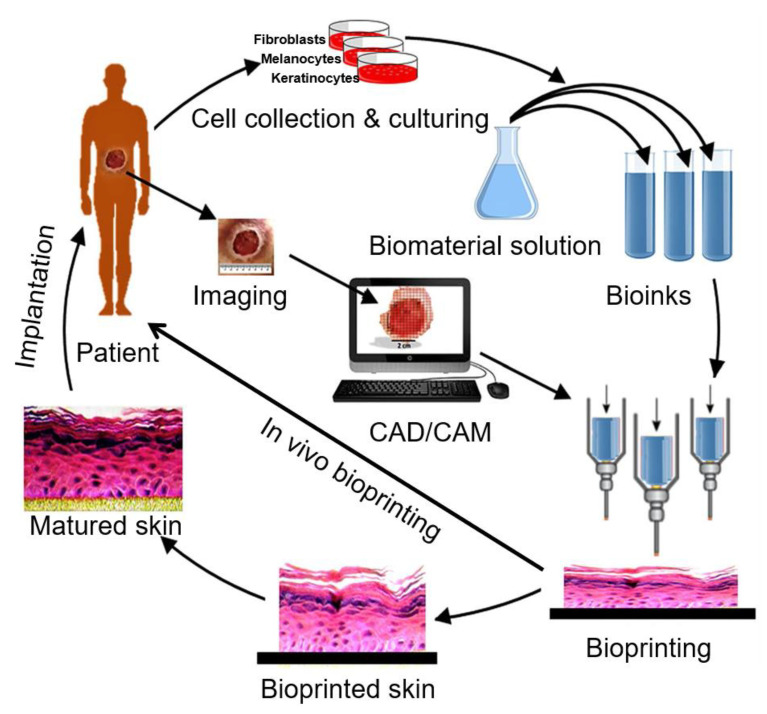
The basic process of 3D bioprinting skin. Various cells like keratinocytes, fibroblasts and melanocytes are collected from the patient and cultured in a cell culture system. A suitable biomaterial is mixed with the cells and the formed bioink is fed to the bioprinting system. Then the skin is printed with appropriate 3D bioprinting technology according to the 3D pattern that is captured from the wound using CAD/CAM approaches. The printed skin could be directly printed to the wound surface or cultured under appropriate conditions to obtain mature skin for transplantation. Adapted from [22], under open access license.

**Figure 3 polymers-12-01237-f003:**
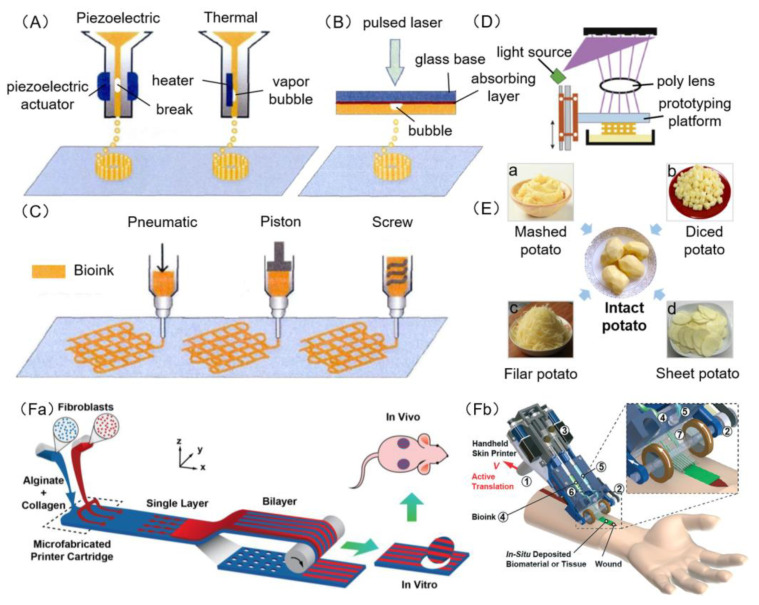
Schematic diagram of 3D bioprinting. (**A**) Piezoelectric inkjet bioprinters form pulses by piezoelectric pressure to force droplets from the nozzle, while thermal bioprinters use air-pressure pulses produced by a printhead that is electrically heated. (**B**) Laser bioprinters use laser focused on an absorbing substrate to generate pressures that propel bioink onto a collector substrate. (**C**) Extrusion bioprinters use pneumatic, piston or screw dispensing systems to extrude continuous beads of bioink. (**D**) Stereolithography bioprinters use a digital light projector to selectively crosslink bioink plane-by-plane. Images (**A**)–(**D**) were adapted with permission from [66]; copyright Journal of Zhejiang University (Engineering Science), 2019. (**E**) Schematic diagram of traditional 3D bioprinting similar to the reverse process of cutting potatoes; laser (**B**), inkjet (**A**), extrusion (**C**), stereolithography (**D**) bioprinting seem like the assembling of potatoes from mashed potato (**a**), diced potato (**b**), filar potato (**c**) and sheet potato (**d**), respectively. (**F**) Microfluidic bioprinters use a microfluidic-based device to extrude a biopolymer sheet with precise spatio-temporal control over the component proportion of bioink. (**a**) Illustration of proposed skin printer for formation of cell-populated skin substitute in a microfabricated printer cartridge and application in vivo. Image (**Fa**) was reproduced with permission from [51]; copyright Chemical and Biological Microsystems Society, 2013. (**b**) Rendered image of handheld bioprinter. ① a handle, ② a stepper motor to define the deposition speed, ③ two on-board syringe pump modules controlling the flow rates of bioink and cross-linker solution, ④ bioink, ⑤ cross-linker solution, ⑥ syringe holder, ⑦ 3D printed microfluidic cartridge for spatial organization of solutions and sheet formation. Image (**Fb**) was reproduced with permission from [65]; copyright Royal Society of Chemistry, 2018.

**Figure 4 polymers-12-01237-f004:**
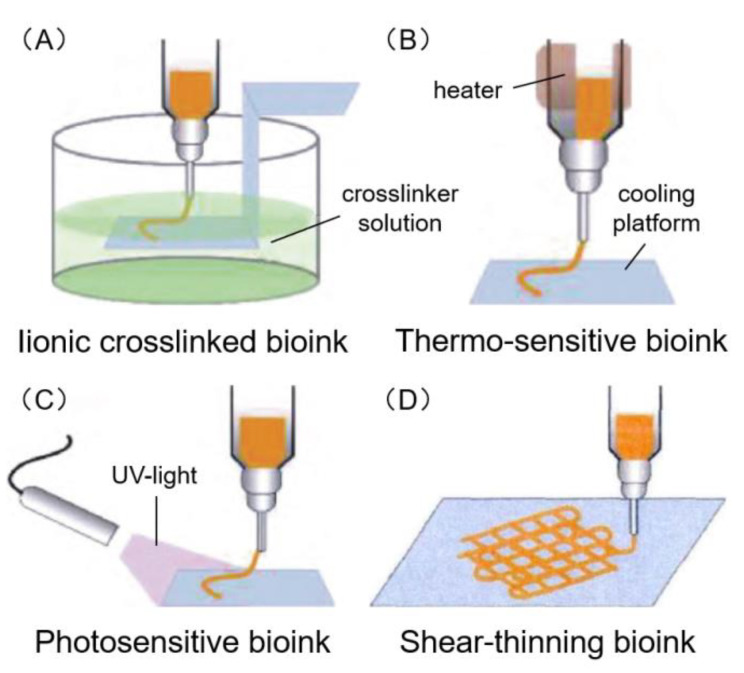
Four common types of bioink. (**A**) Ionic crosslinking bioink mainly performs the solidification through ionic crosslinking reaction. (**B**) Thermo-sensitive bioink mainly performs the transformation from sol to gel state by heating or cooling. (**C**) Photosensitive bioink mainly performs the transformation from sol to gel state by activating the light initiator in bioink. (**D**) Shear-thinning bioink mainly performs the solidification through shear force. The viscosity of some materials decreases with the increase of shear stress, while gelation on the platform occurs with no shear stress. Adapted with permission from [66]; copyright Journal of Zhejiang University (Engineering Science), 2019.

**Figure 5 polymers-12-01237-f005:**
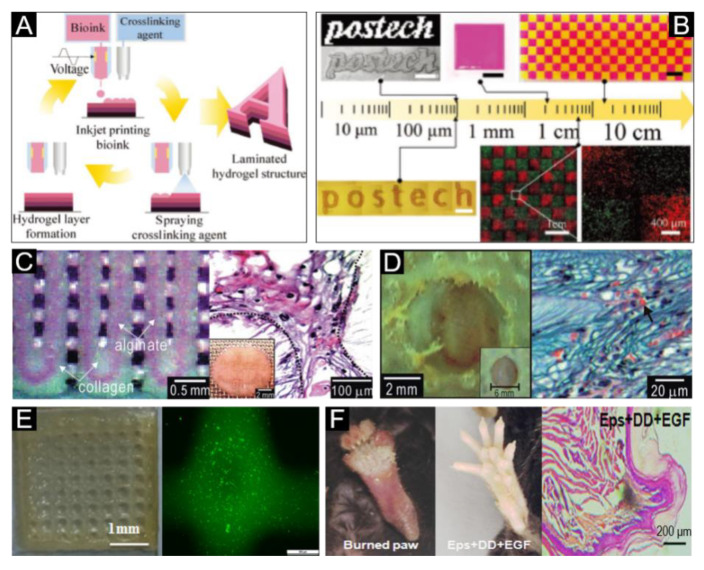
Ionic crosslinked bioink for skin bioprinting. (**A**) Schematic of the inkjet-spray printing process for the fabrication of laminated hydrogel structures. (**B**) Inkjet-spray-printed hydrogel structures of various scales, ranging from several hundreds of micrometers to a few tens of centimeters. The results indicated a high cell viability on the hydrogel structure. Reproduced with permission from [93]; copyright Wiley-VCH, 2018. (**C**) The optical image (Left) and optical cross-section image stained with hematoxylin and eosin (Right) of the fabricated core/shell scaffold. (**D**) The optical image (Left) and histological photomicrograph of the wound cross-section. Arrowheads indicate the generated vasculature. Reproduced with permission from [150]; copyright Royal Society of Chemistry, 2011. (**E**) Representative image (Left) of the cell-laden printed construct and fluorescence image (Right) of cells embedded in the printed construct. (**F**) Sweat gland regeneration (Middle) and histology of wound healing (Right) in mouse paw after 3D-ECM mimics implantation; the burned mouse paw is shown on the left. Reproduced with permission from [82]; copyright Elsevier, 2015.

**Figure 6 polymers-12-01237-f006:**
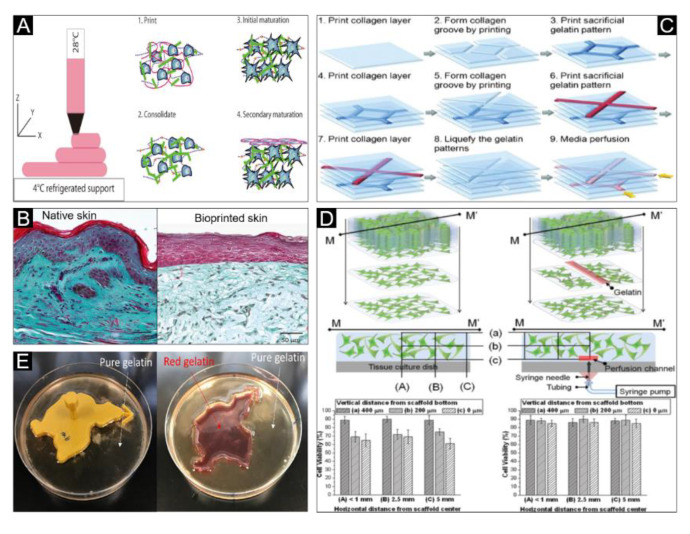
Thermo-sensitive bioink for skin bioprinting. (**A**) Schematic representation of the 3D bioprinting, consolidation and maturation steps using the developed bioink. (**B**) Histological and morphological characterization of native skin and the bioprinted skin. Reproduced with permission from [152]; copyright Wiley-VCH, 2016. (**C**) Schematic procedure of the construction of the multi-layered collagen scaffold with embedding and removal of sacrificial gelatin patterns using the 3D bio-printer. (**D**) Top: Schematics of fibroblast (FB)-laden collagen scaffold construction without (top, left) and with (top, right) embedding and removal of printed sacrificial gelatin channel. Middle: FB viability inspected locations (a vertical section at M–M’) in the collagen scaffolds without (middle, left) and with (middle, right) inside media perfusion. Bottom: Measured FB viability at the inspected locations after 1 week of culture without and with media perfusion. Reproduced with permission from [153]; copyright Wiley Periodicals, 2009. (**E**) Cast pure gelation and red gelatin placed in 4 °C for gelation. Reproduced with permission from [27]; copyright Elsevier, 2018.

**Figure 7 polymers-12-01237-f007:**
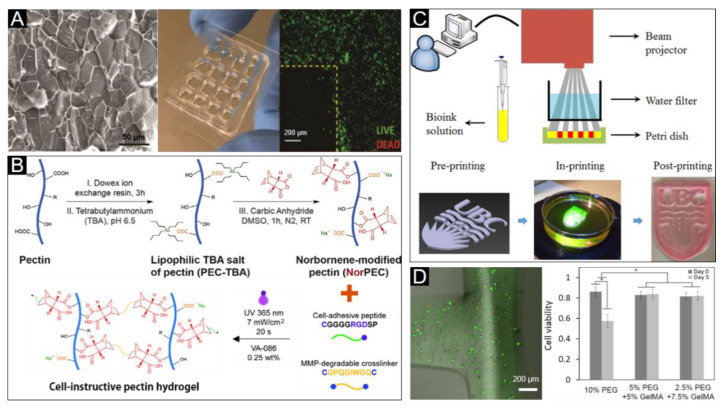
Photosensitive bioink for skin bioprinting. (**A**) Microstructure (left) and image (middle) of the pectin hydrogel with optimal crosslinker, and confocal microscopy image (right) of fibroblasts live/dead staining, which indicated high cell viability embedded within the 3D porous construct. Reproduced with permission from [156]; copyright Royal Society of Chemistry, 2013. (**B**) Schematic illustration of photocrosslinking and biofunctionalization of click NorPEC hydrogels via UV light. Reproduced with permission from [114]; copyright Elsevier, 2017. (**C**) Schematic diagram of developed visible light stereolithography-based bioprinting system and bioprinting procedure of the developed system. (**D**) Qualitative and quantitative cell viability analysis of fibroblasts within the printed structure. Reproduced with permission from [158]; copyright IOP, 2015.

**Figure 8 polymers-12-01237-f008:**
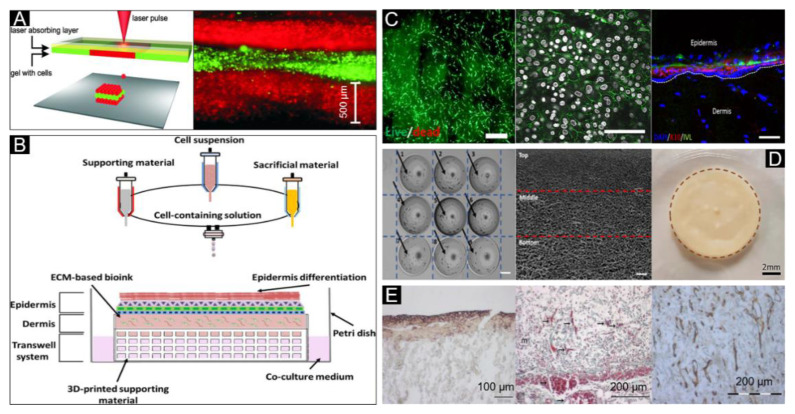
Shear-thinning bioink for skin bioprinting. (**A**) Sketch of the laser printing setup (left) and the alternating color-layers of red (keratinocytes) and green (fibroblasts) (right). Reproduced with permission from [83]; copyright Wiley Periodicals, 2012. (**B**) Extrusion-inkjet printing pattern for fabrication of 3D printed human skin model and its maturation with functional transwell system, all in a single-step process. (**C**) Image and dead images of the contracted compartments at day 7 to reveal their morphology and viability (left, scale bar: 200 μm), immunofluorescence labelling examination of E-cadherin (middle, white: Nuclei, green: E-cadherin, scale bar: 200 μm), and immunofluorescence analysis of the human skin model (Right, DAPI: Nucleus, K10: Early differentiation marker, and IVL: Involucrin, scale bar: 100 μm). Reproduced with permission from [160]; copyright IOP, 2017. (**D**) A two-step 3D bioprinting strategy to manipulate the cell distribution (cells indicated by black arrows—homogeneous cell distribution) (Left), pore size distribution with the 3D collagen matrix-hierarchical porous microstructure (middle), and 3D bioprinted pigmented human skin constructs with uniform skin pigmentation (right), pigmented area is enclosed by the brown dotted line. Reproduced with permission from [116]; copyright IOP, 2018. (**E**) E-cadherin expression (left), blood vessels (arrows) (middle) and collagen IV (right) can be detected in the skin constructs on day 11. Reproduced from [161], under open access license.

**Figure 9 polymers-12-01237-f009:**
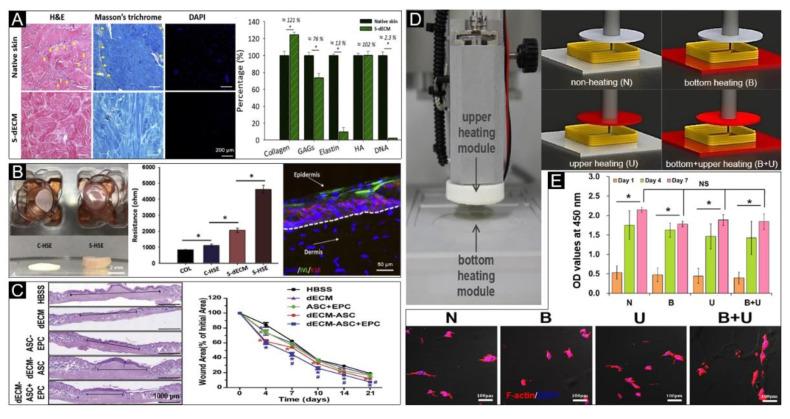
Decellularized extracellular matrix (dECM) bioink for skin bioprinting. (**A**) Qualitative analysis with H&E, Masson’s trichrome and DAPI staining (left) and quantitative analysis (right) of skin-derived dECM bioink, including collagen, GAGs, elastin, hyaluronic acid and DNA, which indicated that it successfully removes the cellular components from the native skin tissue. (**B**) Representative photographs (left) of 3D cell-printed in vitro skin equivalents, electrical resistance values for each group (S-HSE is the group using skin-derived dECM bioink, and the others are control group) (middle) and protein expressions (right) of the S-HSE group on day 10 after ALI culture. Involucrin (IVL): Early epidermal differentiation marker; keratin 10 (K10): Late epidermal differentiation marker. (**C**) Representative photographs of skin wound tissues on day 21 (wound gap, black lines exhibit distances between advancing edges of wounds) (left) and wound areas relative to the orginal ones, indicating the bioprinted skin based on dECM bioink accelerated wound healing. Reproduced with permission from [122]; copyright Elsevier, 2018. (**D**) Photographs of the upper and bottom heating modules installed in 3D printing equipment (left), and conceptual diagram of non-heating and heating conditions (right). (**E**) Qualitative (bottom) and quantitative (upper) analysis of cellular activities under different heating conditions. Reproduced from [23], under open access license.

**Table 1 polymers-12-01237-t001:** Comparison of five types of bioprinting techniques.

Parameters	Inkjet	Laser	Extrusion	Stereolithography	Microfluidic
Printing process	Serial (drop by drop)	Serial (dot by dot)	Serial (line by line)	Parallel and continuous (projection based)	Organ-on-a-chip
Cost	Low	High	Medium	Low	Low
Cell viability	>85%	>95%	40–80%	>85%	>80%
Print speed	Fast	Medium	Slow	Fast	Fast
Supported viscosities	3.5–12 mPa/s	1–300 mPa/s	30–6 × 10^7^ mPa/s	No limitation	0–30 Pa/s
Resolution	High	High	Medium	High	High
Cell density	Low	Medium	High	Medium	High
Representative materials	alginate, collagen	collagen, alginate	alginate, GelMA, collagen	GelMA	collagen, alginate, gelatin
References	[18,30,69,78,79]	[18,69,80,81]	[18,69,82,83]	[69,74,75,84]	[51,65,77,85]

**Table 2 polymers-12-01237-t002:** Crosslinking methods and time of hydrogels commonly used for skin bioprinting.

Hydrogels	Crosslinking Method	Crosslinking Time	References
Collagen	hydrophobic bonding	0.5–60 min	[87,95,96,97]
Gelatin	temperature/glutaraldehyde	minutes to hours	[29,34,98]
Alginate	CaCl_2_	seconds	[36,87,99,100]
Fibrin	thrombin	seconds	[101,102,103,104]
Hyaluronic acid	thiol group crosslink/UV light	15–30 min/s	[35,65,87,105,106]
Chitosan	pH	5–50 min	[87,92,107,108,109]
Pectin	thiol group crosslink	15–30 min	[110,111,112,113,114,116]

**Table 3 polymers-12-01237-t003:** Major growth factors, cytokines and chemokines that participate in wound healing.

Signaling Molecules	Cell	Function
EGF	PLs, MPs, FBs, KCs	Reepithelialization, Angiogenesis [124,125,126]
FGF	KCs, MCs, FBs, ECs, SMCs, CCs	Granulation tissue formation, Reepithelialization, Tissue remodeling [127,128,129]
TGF-β	PLs, KCs, MPs, FBs	Inflammation, Angiogenesis, Granulation tissue formation, Reepithelialization, Tissue remodeling [130,131,132]
PDGF	PLs, KCs, MPs, ECs, FBs	Inflammation, Granulation tissue formation, Angiogenesis, Reepithelialization, Tissue remodeling [133,134]
VEGF	PLs, NPs, MPs, ECs, SMCs, FBs	Inflammation, Angiogenesis, Granulation tissue formation [135]
CTGF	FBs	Granulation tissue formation, Reepithelialization, Tissue remodeling [136]
CM-CSF	KCs	Inflammation, Angiogenesis, Reepithelialization [137]
IL-1	NPs, MOs, MPs, KCs	Inflammation, Reepithelialization [138]
IL-6	NPs, MPs	Inflammation, Reepithelialization [128]
TNF-α	NPs, MPs	Inflammation, Reepithelialization [128]
CXCL1	NPs	Reepithelialization [139]
CXCL8	KCs, LCs	Reepithelialization, Tissue remodeling [139]
CXCL10	KCs	Inflammation [140]
CXCL12	ECs, KCs, MFBs	Inflammation, Angiogenesis, Reepithelialization [140]
CCL2	KCs	Inflammation [141]

PLs: Platelets, MPs: Macrophages, FBs: Fibroblasts, KCs: Keratinocytes, MCs: Mast cells, ECs: Endothelial cells, SMCs: Smooth muscle cells, CCs: Chondrocytes, LCs: Leukocytes, NPs: Neutrophils, MOs: Monocytes, MFBs: Myofibroblasts.

**Table 4 polymers-12-01237-t004:** Materials, cell encapsulation, bio-factors, crosslinker, bioink types, bioprinting technology of the bioink products for skin printing recently.

Materials	Cell Encapsulation	Bio-Factors/Crosslinker	Bioink Type	Bioprinting Technology	Ref.
Alg, Gel	FBs	FBS/CaCl_2_, EDC, NHS	Ionic, thermo-sensitive	Extrusion	[29]
Alg, CNF, GelMA, fibrinogen	FBs	FBS/CaCl_2_, thrombin	Ionic, photosensitive	Inkjet	[93]
Gel, Alg	EPC, DH	FBS, EGF/CaCl_2_	Ionic, thermo-sensitive	Extrusion	[82]
Alg, honey	FBs	FBS/CaCl_2_,	Ionic	Extrusion	[149]
Col, Alg	FBs, KCs	FBS, KGS/CaCl_2_, EDC	Ionic	Extrusion	[150]
Pectin, CS, CD, propolis	FBs	FBS	Ionic	Extrusion	[151]
Gel, Alg	FBs	FBS/CaCl_2_	Thermo-sensitive, ionic	Inkjet	[27]
Fibrinogen, Gel	FBs	FBS/Thrombin	Thermo-sensitive	-	[104]
Gel, Alg, fibrinogen	FBs, KCs	CaCl_2_, thrombin	Thermo-sensitive, ionic	Extrusion	[152]
Col, Gel	FBs	FBS, FGS/NaHCO_3_	Thermo-sensitive	Inkjet	[153]
Fibrinogen, Gel, Gly, HA	KCs, MCs, FBs, pACs, FDPC, DMEC	FBS, EGF, FGF, VEGF, IGF/Thrombin	Thermo-sensitive	Extrusion	[154]
Fibroin, Gel	FBs, KCs	FBS, FGF, EGF/Ty	Thermo-sensitive	Extrusion	[155]
Pectin	FBs, KCs	FBS/CA	Photosensitive	Extrusion	[114]
Pectin	FBs	CaCl_2_, MA	Photosensitive, ionic	Extrusion	[156]
Fibroin, PEG4A	FBs, KCs	FBS/NVP	Photosensitive	Stereolithography	[157]
GelMA, PEG	FBs	FBS/NVP	Photosensitive	Stereolithography	[158]
PEG	FBs, KCs	FBS	Photosensitive	Inkjet	[159]
Col	FBs, KCs	FBS/NaHCO_3_	Shear thinning	Inkjet	[20]
Col	FBs, KCs	FBS, FGS, KGS/NaHCO_3_	Shear thinning	Inkjet	[30]
Col, Alg	FBs, KCs	-	Shear thinning	Microfluidic	[51]
Col, Alg, fibrinogen, HA	FBs, KCs	CaCl_2_, thrombin	Shear thinning, ionic	Microfluidic	[65]
GelMA, Col	FBs, KCs, MCs	FBS/Ty	Shear thinning, photosensitive	Extrusion	[91]
CS, HA, Col	KCs	Ty	Shear thinning	Extrusion	[92]
Col, Alg	FBs, KCs	FBS/CaCl_2_	Shear thinning, ionic	Laser	[83]
Col, Gel	FBs, KCs	FBS, KGS	Shear thinning, thermo-sensitive	Extrusion, inkjet	[160]
Col	FBs, KCs, MCs	FBS, KGS, MGS/NaHCO_3_	Shear thinning	Laser	[79]
Col	FBs, KCs, MCs	FBS, KGS, MGS, FGS	Shear thinning	Inkjet	[116]
Col	FBs, KCs	FBS	Shear thinning	Laser	[161]
Col	FBs, KCs	FBS, FGS, KGS/EDC	Shear thinning	Extrusion	[162]
Col, fibrinogen	AFS, MSCs	FBS/Thrombin	Shear thinning	Extrusion	[94]
dECM	FBs, KCs, ASC, EPC, MNC	FBS, KGS, VEGF, FGF, EGF, IGF	dECM, thermo-sensitive	Inkjet, extrusion	[122]
dECM	FBs	FBS	dECM, thermo-sensitive	Inkjet	[84]
dECM	FBs	FBS	dECM, thermo-sensitive	Extrusion	[23]
dECM	FBs	FBS/EDC	dECM, thermo-sensitive	Extrusion	[81]

Alg: (sodium) Alginate, Gel: Gelatin, GelMA: Gelatin methacrylate, CNF: Cellulose nanofiber, Col: Collagen, CS: Chitosan, CD: Cyclodextrin, HA: Hyaluronic acid, PEG: Polyethylene glycol, PEG4A: Four-arm polyethylene glycol acrylate, Gly: Glycerol, dECM: Decellularized extracellular matrix, FBs: Fibroblasts, KCs: Keratinocytes, MCs: Melanocytes, EPC: Endothelial progenitor cell, DH: Dermal homogenates, pACs: Pre-adipocytes, FDPC: Follicle dermal papillae cell, DMEC: Dermal microvascular endothelial cell, AFS: Amniotic fluid-derived stem, MSCs: Marrow-derived mesenchymal stem cells, MNC: Mononuclear cell, ASC: Adipose mesenchymal stem cell, FBS: Fetal bovine serum, KGS: Keratinocyte growth supplement, FGS: Fibroblast growth supplement, MGS: Melanocyte growth supplement, EGF: Epidermal growth factor, VEGF: Vascular endothelial growth factor, FGF: Fibroblast growth factor, IGF: Insulin-like growth factor, EDC: 1-ethyl-3-(3-dimethyl aminopropyl) carbodiimide, NHS: N-Hydroxysuccinimide, CA: Carbic anhydride, MA: Methacrylic anhydride, NVP: N-vinylpyrrolidone, Ty: Tyrosine.

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
