# Peer review of "Advances in the Research of Bioinks Based on Natural Collagen, Polysaccharide and Their Derivatives for Skin 3D Bioprinting"

_polymers, 2020, doi:10.3390/polym12061237_

Round 1

Reviewer 1 Report

In this review, the author unifies the research advances of bioink in skin bioprinting in recent years. The article content is almost complete and very clear. The extensive knowledge in this review would contribute to who want to understand the recent developments in skin 3D bioprinting. After reviewing this article, it is recommended publishing this article to Polymers.

  1. The article mentioned that current skin 3D bioprinting technology mainly includes inkjet, laser, extrusion, stereolithography and microfluidic bioprinting in the beginning. In the paragraph of classification of bioink, some types of bioink were introduced. It is suggested that the author could describe which bioink is suitable for printing methods in this paragraph.
  2. In order to avoid cell settling in the cell-encapsulating bioink, the viscosity of the bioink is very important. Is there more literature discussing this?

Author Response

Response to Reviewer 1 Comments

Point 1: The article mentioned that current skin 3D bioprinting technology mainly includes inkjet, laser, extrusion, stereolithography and microfluidic bioprinting in the beginning. In the paragraph of classification of bioink, some types of bioink were introduced. It is suggested that the author could describe which bioink is suitable for printing methods in this paragraph.

Response 1: Many thanks for the suggestion of reviewer. Following the suggestion of the reviewer, we added the sentence that “As shown in Table 4, the ionic crosslinked bioink was suitable for extrusion, inkjet, laser, and microfluidic bioprinting” in line 512 (section 6.1), the sentence that “As shown in Table 4, the thermo-sensitive bioink was suitable for extrusion and inkjet bioprinting” in line 550 (section 6.2), the sentence that “As shown in Table 4, the photosensitive bioink was suitable for stereolithography, extrusion, and inkjet bioprinting” in line 579 (section 6.3), the sentence that “As shown in Table 4, the shear-thinning bioink was suitable for extrusion, inkjet, laser, and microfluidic bioprinting” in line 613 (section 6.4), and the sentence that “As shown in Table 4, the dECM bioink was suitable for extrusion and inkjet bioprinting” in line 667 (section 6.5).

Point 2: In order to avoid cell settling in the cell-encapsulating bioink, the viscosity of the bioink is very important. Is there more literature discussing this?

Response 2: Many thanks for the suggestion of reviewer. Following the suggestion of the reviewer, we added two other references, [147] and [148] in line 480 (section 5.2). Wang et al. [148] had declared that the viscosity of hydrogel can affect cell behavior including cell morphology, proliferation, which indicating viscosity of the bioink can affect cell density onto 3D hydrogel structure, even resulting cell settling. Nichol et al. [147] fabricated a cell-laden gelatin methacrylate (GelMA) hydrogel and cultured at optimal condition, the result also demonstrated that cell density significantly onto hydrogel increased with increased GelMA concentration, which indicating the viscosity of the bioink is very important to cell behavior.

Reviewer 2 Report

The purpose of this review is to bring a perspective about the 3D bioprinting in Skin Tissue Engineering and also provide insights regarding the development of Bioink and printing technique for that application.

It is an interesting topic and the authors found a good way to write about it in the manuscript. However, there are suggestions regarding the structure and some information the authors could add to enrich the quality of the publication.

1)The structure and function of the skin are known to everyone and it diverts the interest of the reader from knowing about the advancement of printing techniques and bioink for skin regeneration. I recommend to remove the section of skin function and wound injury. Instead, try to include a paragraph giving emphasis to the prospective researches in the field and the possible contribution to wound healing.

2)Along with developed vasculature, regeneration of nerves/sensation is a critical challenge. Hence the author needs to discuss that too. 

Author Response

Response to Reviewer 2 Comments

Point 1: The structure and function of the skin are known to everyone and it diverts the interest of the reader from knowing about the advancement of printing techniques and bioink for skin regeneration. I recommend to remove the section of skin function and wound injury. Instead, try to include a paragraph giving emphasis to the prospective researches in the field and the possible contribution to wound healing.

Response 1: Many thanks for the suggestion of reviewer. Following the suggestion of the reviewer, we removed the section of skin structure and function, and revised to “The skin, the largest organ of the human body, accounting for about 15% of the total body weight in adults, has a very complex multi-layered structure, including epidermis, dermis and hypodermis. It plays a critical role in maintaining homeostasis, temperature regulation, metabolite transportation, sensory perception, especially acts as an important barrier against the external environment, preventing the invasion of pathogenic microorganisms” in line 92 (the 1th paragraph, section 2).

Due to the remove of the section, the references [43-51] of the last manuscript have been removed, and the serial number of the other references had been moved forward in order; the title of section 2 was revised to “Skin Damage and Wound Repair”; the serial number of the other figure had been moved forward in order owning to the Figure 2 in the last manuscript was removed. The sentence that “In this review, we outline the structure and function of skin, the current skin bioprinting technologies and the bioink components for skin bioprinting” in abstract was revised to “In this review, we outline the current skin bioprinting technologies and the bioink components for skin bioprinting”, and the comment that “we briefly describe the structure and function of human skin and discuss 3D bioprinting technologies to repair skin injuries” in the first sentence of last paragraph in introduction section has been removed.

And “the prospective researches in the field and the possible contribution to wound healing” has been introduced in penultimate paragraph of section 7, Discussion and Future Perspective.

Point 2: Along with developed vasculature, regeneration of nerves/sensation is a critical challenge. Hence the author needs to discuss that too.

Response 2: Many thanks for the suggestion of reviewer. As adnexal structures of human skin, melanin pigmentation, sweat gland, hair follicles, sebaceous gland, and sensory nerve, they are responsible for protection against from ultraviolet radiation, temperature regulation, preserve moisture and sensory functions. Therefore, as wound healing involves the functional recovery of skin structures, bioprinted skin are also expected to support the development of sweat glands, hair follicles, sebaceous glands, sensory nerve and melanin pigmentation. Following the suggestion of the reviewer, we added “, sensory nerve” in line 770 (the 5th paragraph, section 7), revised the sentence that “……involving sweat glands, hair follicles, sebaceous glands for wound healing” in line 789 (the 6th paragraph, section 7) to “……involving sebaceous glands, sweat glands, hair follicles, melanin pigmentation and sensory nerve for wound healing”.
